# Fast Rates for Bandit Optimization with Upper-Confidence Frank-Wolfe

**Quentin Berthet** [*]
University of Cambridge
q.berthet@statslab.cam.ac.uk

**Vianney Perchet** [†]
ENS Paris-Saclay & Criteo Research, Paris
vianney.perchet@normalesup.org

## Abstract

We consider the problem of bandit optimization, inspired by stochastic optimization and online learning problems with bandit feedback. In this problem, the objective is to minimize a global loss function of all the actions, not necessarily a cumulative loss. This framework allows us to study a very general class of problems, with applications in statistics, machine learning, and other fields. To solve this problem, we analyze the Upper-Confidence Frank-Wolfe algorithm, inspired by techniques for bandits and convex optimization. We give theoretical guarantees for the performance of this algorithm over various classes of functions, and discuss the optimality of these results.

## Introduction

In online optimization problems, a decision maker choses at each round $t \geq 1$ an action $\pi_t$ from some given action space, observes some information through a feedback mechanism in order to minimize a loss, function of the set of actions $\{\pi_1, \ldots, \pi_T\}$. Traditionally, this objective is computed as a cumulative loss of the form $\sum_t \ell_t(\pi_t)$ [20, 34], or as a function thereof [2, 3, 16, 32].

Examples include classical multi-armed bandit problems where the action space is finite with $K$ elements, in stochastic or adversarial settings [9]. In these problems, the loss at round $t$ can be written as $\ell_t(e_{\pi_t})$ for a linear form $\ell_t$ on $\mathbf{R}^K$, and basis vectors $e_i$. More generally, this includes also bandit problems over a convex body $\mathcal{C}$, where the action at each round consists in picking $x_t \in \mathcal{C}$ and where the loss $\ell_t(x_t)$ is for some convex function $\ell_t$ [see, e.g. 9, 12, 19, 10].

In this work, we consider the online learning problem of *bandit optimization*. Similarly to other problems of this type, a decision maker chooses at each round an action $\pi_t$ from a set of size $K$, and observes information about an unknown convex loss function $L$. The difference is that the objective is to minimize a global convex loss $L\left(\frac{1}{T} \sum_{t=1}^{T} e_{\pi_t}\right)$, not a cumulative one. At each round, choosing the $i$-th action increases the information about the local dependency of $L$ on its $i$-th coefficient. This problem can be contrasted with the objective of minimizing the average pseudo-regret in a stochastic bandit problem, i.e. of minimizing $\frac{1}{T} \sum_{t=1}^{T} L(e_{\pi_t})$ with observation $\ell_t(e_{\pi_t})$, a noisy estimate of $L(e_{\pi_t})$. At the intersection of these frameworks, when $L$ is a linear form, is the stochastic multi-armed bandit problem. Our problem is also related to maximization of known convex objectives [2, 3]. We compare our framework to these settings in Section 1.4.

Bandit optimization shares some similarities with stochastic optimization problems, where the objective is to minimize $f(x_T)$ for an unknown function $f$, while choosing at each round a variable $x_t$

---

[*]Supported by an Isaac Newton Trust Early Career Support Scheme and by The Alan Turing Institute under the EPSRC grant EP/N510129/1.

[†]Supported by the ANR (grant ANR- 13-JS01-0004-01), and the *FMJH Program Gaspard Monge in Optimization and operations research* (supported in part by EDF) and from the Labex LMH.

and observing some noisy information about the function $f$. Our problem can be seen as a stochastic optimization problem over the simplex, with the caveat that the list of actions $\pi_1, \ldots, \pi_T$ determines the variable, as $x_t = \frac{1}{t} \sum_{s=1}^{t} e_{\pi_s}$, as well as the manner in which additional information about the function can be gathered. This setting allows us to study a more general class of problems than multi-armed bandits, and to cover examples where there is not one optimal action, but rather an optimal global strategy, that is an optimal mix of actions. We describe several natural problems from machine learning, statistics, or economics that are cases of bandit optimization.

This problem draws inspiration from the world of multi-armed bandit problems and that of stochastic convex optimization, and our solution to it does as well. We analyze the Upper-Confidence Frank-Wolfe algorithm, a modification of the Frank-Wolfe algorithm [17] and of the UCB algorithm for bandits [5]. The link with Frank-Wolfe is related to the choice of one action, and encourages exploitation, while the link with UCB encourages to chose rarely picked actions in order to increase knowledge about the function, encouraging exploration. This algorithm can be used for all convex functions $L$, and performs in a near-optimal manner over various classes of functions. Indeed, if it has been already proved that it achieves slow rates of convergence in some cases, i.e., the error decreases as $1/\sqrt{T}$, we are able to exhibit fast rates decreasing in $1/T$, up to logarithmic terms.

These fast rates are surprising, as they sometimes even hold for non-strongly convex functions, and in many problems with bandit feedback they cannot be reached [23, 35]. As shown in our lower bounds, the main complexity of this problem is statistical and comes from the limited information available about the unknown function $L$. Usual results in optimization with a known function are not necessarily relevant to our problem. As an example, while linear rates in $e^{-cT}$ are possible in deterministic settings with variants in the Frank-Wolfe algorithm, we are limited to fast rates in $1/T$ under similar assumptions. Interestingly, while linear functions are one of the settings in which the deterministic Frank-Wolfe algorithm is the most efficient, it is among the most complicated for bandit optimization, and only slow rates are possible (see theorems 2 and 6).

Our work is organized in the following manner: we describe in Section 1 the problem of bandit optimization. The main algorithm is introduced in Section 2, and its performance in various settings is studied in Section 3, 4, and 5. All proofs of the main results are in the supplementary material.

**Notations:** For any positive integer $n$, denote by $[n]$ the set $\{1, \ldots, n\}$ and, for any positive integer $K$, by $\Delta_K := \left\{ p \in \mathbf{R}^K : p_i \geq 0 \text{ and } \sum_{i \in [K]} p_i = 1 \right\}$ the unit simplex of $\mathbf{R}^K$. Finally, $e_i$ stands for the $i$-th vector of the canonical basis of $\mathbf{R}^K$. Notice that $\Delta_K$ is their convex hull.

# 1 Bandit Optimization

We describe the *bandit optimization* problem, generalizing multi-armed bandits. This stochastic optimization problem is doubly related to bandits: The decision variable cannot be chosen freely but is tied to the past actions, and information about the function is obtained via a bandit feedback.

## 1.1 Problem description

A each time step $t \geq 1$, a decision maker chooses an action $\pi_t \in [K]$ from $K$ different actions with the objective of minimizing an unknown convex loss function $L : \Delta_K \to \mathbf{R}$. Unlike in traditional online learning problems, we do not assume that the overall objective of the agent is to minimize a cumulative loss $\sum_t L(e_{\pi_t})$ but rather to minimize the global loss $L(p_T)$, where $p_t \in \Delta_K$ is the vector of *proportions* of each action (also called occupation measure), i.e.,

$$p_t = \left( T_1(t)/t, \ldots, T_K(t)/t \right) \text{ with } T_i(t) = \sum_{s=1}^{t} \mathbf{1}\{\pi_s = i\} .$$

Alternatively, $p_t = \frac{1}{t} \sum_{i=1}^{t} e_{\pi_s}$. As usual in stochastic optimization, the performance of a policy is evaluated by controlling the difference

$$r(T) := \mathbf{E}[L(p_T)] - \min_{p \in \Delta_K} L(p) .$$

The information available to the policy is a feedback of *bandit type*: given the choice $\pi_t = i$, it is an estimate $\hat{g}_t$ of $\nabla L(p_t)$. Its precision, with respect to each coefficient $i \in [K]$, is specified by a deviation function $\alpha_{t,i}$, meaning that for all $\delta \in (0, 1)$, it holds with probability $1 - \delta$ that

$$|\hat{g}_{t,i} - \nabla_i L(p_t)| \leq \alpha_{t,i}(T_i(t), \delta) .$$

At each round, it is possible to improve the precision for one of the coefficients of the gradient but possibly at a cost of increasing the global loss. The most typical case, described in the following section, is of $\alpha_{t,i}(T_i, \delta) = \sqrt{2 \log(t/\delta)/T_i}$, when the information consists of observations from different distributions. In general, this type of feedback mechanism is indicative of a bandit feedback (and not of a full information setting), as motivated by the following parametric setting.

## 1.2 Bandit feedback and parametric setting

One of the motivations is the minimization of a loss function $L$ belonging to a known class $\{L(\mu, \cdot), \mu \in \mathbf{R}^K\}$ with an unknown parameter $\mu$. Choosing the $i$-th action provides information about $\mu_i$, through an observation of some auxiliary distribution $\nu_i$.

As an example, the classical stochastic multi-armed bandit problem [9] falls within our framework. Denoting by $\mu_i$ the expected loss of arm $i \in [K]$, the average pseudo-regret $\bar{R}$ can be expressed as

$$\bar{R}(t) = \frac{1}{t} \sum_{s=1}^{t} \mu_{\pi_s} - \mu^\star = \sum_{i=1}^{K} \mu_i \frac{T_i(t)}{t} - \mu^\star = p_t^\top \mu - p^{\star\top} \mu, \quad \text{with } p^\star = e_{i^\star},$$

Hence the choice of $L(\mu, p) = \mu^\top p$ corresponds the problem of multi-armed bandits. Since $\nabla L(\mu, p) = \mu$, the feedback mechanism for $\hat{g}_t$ is induced by having a sample $X_t$ from $\nu_{\pi_t}$ at time step $t$, taking $\hat{g}_{t,i} = \bar{X}_{t,i}$, the empirical mean of the $T_i(t)$ observations $\nu_i$. In this case, if $\nu_i$ is sub-Gaussian with parameter 1, we have $\alpha_{t,i}(T_i, \delta) = 2\sqrt{2 \log(t/\delta)/T_i}$.

More generally, for any parametric model, we can consider the following observation setting: For all $i \in [K]$, let $\nu_i$ be a sub-Gaussian distribution with mean $\mu_i$ and tail parameter $\sigma^2$. At time $t$, for an action $\pi_t \in [K]$, we observe a realization from $\nu_{\pi_t}$. We estimate $\mu_i$ by the empirical mean $\hat{\mu}_{t,i}$ of the $T_i(t)$ draws from $\nu_i$, and $\hat{g}_t = \nabla_p L(\hat{\mu}_t, p_t)$ as an estimate of the gradient of $L = L(\mu, \cdot)$ at $p_t$. The following bound on $\alpha_i$ under smoothness conditions on the parametric model is a direct application of Hoeffding's inequality.

**Proposition 1.** *Let $L = L(\mu, \cdot)$ for some $\mu \in \mathbf{R}^K$ being $\mu$-gradient-Lipschitz, i.e., such that*

$$\left| \left( \nabla_p L(\mu, p) \right)_i - \left( \nabla_p L(\mu', p) \right)_i \right| \leq |\mu_i - \mu'_i| \ , \ \forall p \in \Delta([K]).$$

*Under the sub-Gaussian observation setting above, $\hat{g}_t = \nabla_p L(\hat{\mu}_t, p_t)$ is a valid gradient feedback with deviation bounds $\alpha_{t,i}(T_i, \delta) = \sqrt{2\sigma^2 \log(t/\delta)/T_i}$.*

This Lipschitz condition on the parameter $\mu$ gives a motivation for our gradient bandit feedback.

## 1.3 Examples

**Stochastic multi-armed bandit:**

As noted above, the stochastic multi-armed bandit problem is a special case of our setting for a loss $L(p) = \mu^\top p$, and the bandit feedback allows to construct a proxy for the gradient $\hat{g}_t$ with deviations $\alpha_i$ decaying in $1/\sqrt{T_i}$. The UCB algorithm used to solve this problem inspires our algorithm that generalizes to any loss function $L$, as discussed in Section 2.

**Online experimental design:** In the context of statistical estimation with heterogenous data sources [8], consider the problem of allocating samples in order to minimize the variance of the final estimate. At time $t$, it is possible to sample from one of $K$ distributions $\mathcal{N}(\theta_i, \sigma_i^2)$ for $i \in [K]$, the objective being to minimize the average variance of the simple unbiased estimator

$$\mathbf{E}[\|\hat{\theta} - \theta\|_2^2] = \sum_{i \in [K]} \sigma_i^2 / T_i \quad \text{equivalent to} \quad L(p) = \sum_{i \in [K]} \sigma_i^2 / p_i \ .$$

For unknown $\sigma_i$, this problem falls within our framework and the gradient with coordinates $-\sigma_i^2/p_i^2$ can be estimated by using the $T_i$ draws from $\mathcal{N}(\theta_i, \sigma_i^2)$ to construct $\hat{\sigma}_i^2$. This function is only defined on the interior of the simplex and is unbounded, matters that we discuss further in Section 4.3. Other objective functions than the expected $\ell_2$ norm of the error can be used, as in [11], who consider the $\ell_\infty$ norm of the actual estimated deviations, not its expectation.

**Utility maximization:** A classical model to describe the utility of an agent purchasing $x_i$ units of $K$ different goods is the Cobb-Douglas utility (see e.g. [27]) defined for parameters $\beta_i \in (0, 1)$ by

$$U(x_1, \ldots, x_K) = \prod_{i \in [K]} x_i^{\beta_i} .$$

Maximizing this utility for unknown $\beta_i$ under a budget constraint - where each price is assumed to be 1 for ease of notations - by buying one unit of one of $K$ goods at each round, is therefore equivalent to minimizing in $p_i$ (the proportion of good $i$ in the basket) $L(p) = -\sum_{i \in [K]} \beta_i \log(p_i)$.

**Other examples:** More generally, the notion of bandit optimization can be applied to any situation where one optimizes a strategy through actions that are taken sequentially, with information gained at each round, and where the objective depends only on the proportions of actions. Other examples include a problem inspired by online Markovitz portfolio optimization, where the goal is to minimize $L(p) = p^\top \Sigma p - \lambda \mu^\top p$, with a known covariance matrix $\Sigma$ and unknown returns $\mu$, or several generalizations of bandit problems such as minimizing $L(p) = \sum_{i \in [K]} f_i(\mu_i) p_i$ when observations are drawn from a distribution with mean $\mu_i$, for known $f_i$.

### 1.4 Comparison with other problems

As mentioned in the introduction, the problem of bandit optimization is different from online learning problems related to regret minimization [21, 1, 10], even in a stochastic setting. While the usual objective is to minimize a cumulative regret related to $\frac{1}{T} \sum_t \ell_t(x_t)$, we focus on $L(\frac{1}{T} \sum_t e_{\pi_t})$.

Problems related to online optimization of global costs or objectives have been studied in similar settings [2, 3, 16, 32]. They are equivalent to minimizing a loss $L(p_T^\top V)$ where $V$ is a $K \times d$ unknown matrix and $L(\cdot) : \mathbf{R}^d \to \mathbf{R}$ is known. The feedback at stage $t$ is a noisy evaluations of $V_{\pi_t}$. In the stochastic case [2, 3], this is close to our setting - even though none of them subsumes directly the other one. Only slow rates of convergence of order $1/\sqrt{T}$ are derived for the variant of Frank-Wolfe, while we aim at fast rates, which are optimal. In contrast, in the adversarial case [16, 32], there are instances of the problem where the average regret cannot decrease to zero [26].

Using the Frank-Wolfe algorithm in a stochastic optimization problem has also already been considered, particularly in [25], where the estimates of the gradients are increasingly precise in $t$, independently of the actions of the decision maker. This setting, where the action at each round is to pick $x_t$ in the domain in order to minimize $f(x_T)$ is therefore closer to classical stochastic optimization than online learning problems related to bandits [9, 19, 10].

## 2 Upper-Confidence Frank-Wolfe algorithm

With linear functions, as in multi-armed bandits, an estimate of the gradient can be established by using the past observations, as well as confidence intervals on each coefficient in $1/\sqrt{T_i}$. The UCB algorithm instructs to pick the action with the smallest lower confidence estimate $\underline{\mu}_{t,i}$ for the loss. This is equivalent to making a step of size $1/(t+1)$ in the direction of the corner of the simplex $e$ that minimizes $e^\top \underline{\mu}_t$. Following this intuition, we introduce the UCB Frank-Wolfe algorithm that uses a proxy of the gradient, penalized by the size of confidence intervals.

---

**Algorithm 0: UCB Frank-Wolfe algorithm**

**Input**: $K$, $p_0 = \mathbf{1}_{[K]}/K$, sequence $(\delta_t)_{t \geq 0}$;
**for** $t \geq 0$ **do**
    Observe $\hat{g}_t$, noisy estimate of $\nabla L(p_t)$;
    **for** $i \in [K]$ **do**
        $\hat{U}_{t,i} = \hat{g}_{t_i} - \alpha_{t,i}(T_i(t), \delta_t)$
    **end**
    Select $\pi_{t+1} \in \operatorname{argmin}_{i \in [K]} \hat{U}_{t,i}$;
    Update $p_{t+1} = p_t + \frac{1}{t+1}(e_{\pi_{t+1}} - p_t)$
**end**

---

Notice that for any algorithm, the selection of an action $\pi_{t+1} \in [K]$ at time step $t + 1$ updates the variable $p$ with respect to the following dynamics

$$p_{t+1} = \left(1 - \frac{1}{t+1}\right) p_t + \frac{1}{t+1} e_{\pi_{t+1}} = p_t + \frac{1}{t+1} (e_{\pi_{t+1}} - p_t). \qquad (1)$$

This is implied by the mechanism of the problem, and is not dependent on the choice of an algorithm. If the choice of $e_{\pi_{t+1}}$ is $e_{\star t+1}$, the minimizer of $s^\top \nabla L(p_t)$ over all $s \in \Delta_K$, this would precisely be the Frank-Wolfe algorithm with step size $1/(t + 1)$. Inspired by this similarity, our selection rule is driven by the same principle, using a proxy $\hat{U}_t$ for $\nabla L(p_t)$ based on the information up to time $t$. Our selection rule is therefore driven by two principles, borrowing from tools in convex optimization (the Frank-Wolfe algorithm) and classical bandit problems (Upper-confidence bounds).

The choice of action $\pi_{t+1}$ is equivalent to taking $e_{\pi_{t+1}} \in \operatorname{argmin}_{s \in \Delta_K} s^\top \hat{U}_t$. The computational cost of this procedure is very light, and apart from gradient computations, it is linear in $K$ at each iteration, leading to a global cost of order $KT$.

## 3 Slow rates

In this section we show that when $\alpha_i$ is of order $1/\sqrt{T_i}$, as motivated by the parametric model of Section 1.2, our algorithm has an approximation error of order $\sqrt{\log(T)/T}$ over the very general class of smooth convex functions. We refer to this as the *slow rate*. Our analysis is based on the classical study of the Frank-Wolfe algorithm [see, e.g. 22, and references therein]. We consider the case of $C$-smooth convex functions on the unit simplex, for which we recall the definition.

**Definition** (Smooth functions). For a set $\mathcal{D} \subset \mathbf{R}^n$, a function $f : \mathcal{D} \to \mathbf{R}$ is said to be a $C$-smooth function if it is differentiable and if its gradient is $C$-Lipshitz continuous, i.e. the following holds

$$\|\nabla f(x) - \nabla f(y)\|_2 \leq C \|x - y\|_2 \ , \ \forall x, y \in \mathcal{D} \,.$$

We denote by $\mathcal{F}_{C,K}$ the set of $C$-smooth convex functions. They attain their minimum at a point $p_\star \in \Delta_K$ and their Hessian is uniformly bounded, $\nabla^2 L(p) \preceq C I_K$, if they are twice differentiable. We establish in this general setting a slow rate when $\alpha_i$ decreases like $1/\sqrt{T_i}$.

**Theorem 2** (Slow rate). *Let $L$ be a $C$-smooth convex function over the unit simplex $\Delta_K$. For any $T \geq 1$, after $T$ steps of the UCB Frank-Wolfe algorithm with a bandit feedback such that $\alpha_{t,i}(T_i, \delta) = 2\sqrt{\log(t/\delta)/T_i}$ and the choice $\delta_t = 1/t^2$, it holds that*

$$\mathbf{E}\big[L(p_T)\big] - L(p_\star) \leq 4\sqrt{\frac{3K \log(T)}{T}} + \frac{C \log(eT)}{T} + \left(\frac{\pi^2}{6} + K\right) \frac{2\|\nabla L\|_\infty + \|L\|_\infty}{T}.$$

The proof draws inspiration from the analysis of the Frank-Wolfe algorithm with stepsize of $1/(t+1)$ and of the UCB algorithm. Notice that our algorithm is adaptive to the gradient Lipschitz constant $C$, and that the leading term of the error does not depend on it. We also emphasize the fact that the dependency in $\sqrt{K}$ is expected, and optimal, in bandit setting.

For linear mappings $L(p) = p^\top \mu$, our analysis is equivalent to studying the UCB algorithm in multi-armed bandits. The slow rate in Theorem 2 corresponds to a regret of order $\sqrt{KT \log(T)}$, the distribution-independent (or worst case) performance of UCB. The extra dependency in $\sqrt{\log(T)}$ could be reduced to $\sqrt{\log(K)}$ or even optimally to 1 by using confidence intervals more carefully tailored, for instance by replacing the $\log(t)$ term appearing in the definition of the estimated gradients by $\log(T/T_i(t))$ or $\log(T/KT_i(t))$ if the horizon $T$ is known in advance as in the algorithms MOSS or ETC (see [4, 29, 30]), but at the cost of a more involved analysis.

Thus, multi-armed bandits provide a lower bound for the approximation error $\mathbf{E}[L(p_T)] - L(p_\star)$ of order $\sqrt{K/T}$ for smooth convex functions. We discuss lower bounds further in Section 5.

For the sake of clarity, we state all our results when $\alpha_{t,i}(T_i, \delta) = 2\sqrt{\log(t/\delta)/T_i}$, but our techniques handle more general deviations as $\alpha_{t,i}(T_i, \delta) = \big(\theta \log(t/\delta)/T_i\big)^\beta$ where $\theta \in \mathbf{R}$ and $\beta > 0$ are some known parameters. More general results can be found in the supplementary material.

# 4 Fast rates

In this section, we describe situations where the approximation error rate can be improved to a *fast rate* of order $\log(T)/T$, when we consider various classes of functions, with additional assumptions.

## 4.1 Stochastic multi-armed bandits and functions minimized on vertices

A very natural and well-known - yet illustrative - example of such a restricted class of functions is simply the case of classical bandits where $\Delta^{(i)} := \mu_i - \mu_\star$ is bounded away from 0 for $i \neq \star$. Our analysis of the algorithm can be adapted to this special case with the following result.

**Proposition 3.** *Let $L$ be the linear function $p \mapsto p^\top \mu$. After $T$ steps of the UCB Frank-Wolfe algorithm with a bandit feedback such that $\alpha_{t,i}(T_i, \delta) = 2\sqrt{\log(t/\delta)/T_i}$, the choices of $\delta_t = 1/t^2$ hold the following*

$$\mathbf{E}[L(p_T)] - L(p_\star) \leq \frac{48 \log(T)}{T} \sum_{i \neq \star} \frac{1}{\Delta^{(i)}} + 3\Big(\frac{\pi^2}{3} + K\Big) \frac{\sqrt{K}\|\mu\|_\infty}{T} \ .$$

The constants of this proposition are sub-optimal (for instance the 48 can be reduced up to 2 using more careful but involved analysis). It is provided here to show that this classical bound on the pseudo-regret in stochastic multi-armed bandits [see e.g. 9, and references therein] can be recovered with Frank-Wolfe type of techniques illustrating further the links between bandit problems and convex optimization [20, 34]. This result can actually be generalized to any convex functions which is minimized on a vertex of the simplex with a gradient whose component-wise differences are bounded away from 0.

**Proposition 4.** *Let $L$ be a convex mapping that attains its minimum on $\Delta_K$ at a vertex $p^\star = e_{i^\star}$ and such that $\Delta^{(i)}(L) := \nabla_i L(p^\star) - \nabla_{i^\star} L(p^\star) > 0$ for all $i \neq i^\star$. Then, after $T$ steps of the UCB Frank-Wolfe algorithm with a bandit feedback such that $\alpha_{t,i}(T_i, \delta) = 2\sqrt{\log(t/\delta)/T_i}$, the choices of $\delta_t = 1/t^2$ hold the following*

$$\mathbf{E}[L(p_T)] - L(p_\star) \leq \rho(L)\Big(\frac{48 \log(T)}{T} \sum_{i \neq \star} \frac{1}{\Delta^{(i)}(L)} + \frac{C \log(eT)}{T} + \big(\frac{\pi^2}{6} + K\big)\frac{2\|\nabla L\|_\infty + \|L\|_\infty}{T}\Big),$$

*where $\rho(L) = \Big(1 + \frac{CK}{\Delta_{\min}(L)}\Big)$ and $\Delta_{\min}(L) = \min_{i \neq i_\star} \Delta^{(i)}(L)$.*

The KKT conditions imply that $\Delta^{(i)}(L) \geq 0$ whenever $p^\star$ is in a corner, but the strict inequality is not always guaranteed. In particular, this result may not hold if $p^\star$ is the global minimum of $L$ over $\mathbf{R}^K$. This type of condition has also been linked with rates of convergence in stochastic optimization problems [15].

The extra multiplicative factor $\rho(L)$ can be large, but it would be of the order of $1 + o(1)$ using variants of our algorithms with results that holds only with great probability (typically with confidence bounds of the form $2\sqrt{\log(1/\delta)/T_i}$).

## 4.2 Strongly convex functions

Another classical assumption in convex optimization is strong convexity, as recalled below. We denote by $\mathcal{S}_{\mu,K}$ the set of $\mu$-strongly convex functions of $\Delta_K$. This assumption usually improves the rates in errors of approximation in many settings, even in stochastic optimization or some settings of online learning [see, e.g. 31, 14, 33, 6, 18, 19, 7]. Interestingly enough though, strong convexity cannot be leveraged to improve rates of convergence in online convex optimization [35, 23], where the $1/\sqrt{T}$ rate of convergence cannot be improved. Moreover, leveraging strong convexity usually implies to adapt step size of gradient descents or with linear search and/or away steps for classical Frank-Wolfe methods. Those techniques cannot be adapted to our setting where step sizes are fixed.

**Definition** (Strongly convex functions). . For a set $\mathcal{D} \subset \mathbf{R}^n$, a function $f : \mathcal{D} \to \mathbf{R}$ is said to be a $\mu$-strongly convex if for all $x, y \in \mathcal{D}$, we have

$$f(x) \geq f(y) + \nabla f(x)^\top (x - y) + \frac{\mu}{2}\|x - y\|_2^2 \ .$$

We already covered the case where the convex functions are minimized outside the simplex. We will now assume that the minimum lies in its relative interior.

**Theorem 5.** *Let* $L : \Delta_K \to \mathbf{R}$ *be a C-smooth, $\mu$-strongly convex function such that its minimum $p_\star$ satisfies* $\mathrm{dist}(p_\star, \partial\Delta_K) \geq \eta$, *for some* $\eta \in (0, 1/K]$. *After $T$ steps of the UCB Frank-Wolfe algorithm with a bandit feedback such that* $\alpha_{t,i}(T_i, \delta) = 2\sqrt{\log(t/\delta)/T_i}$, *it holds that, with the choice of* $\delta_t = 1/t^2$,

$$\mathbf{E}[L(p_T)] - L(p_\star) \leq c_1 \frac{\log^2(T)}{T} + c_2 \frac{\log(T)}{T} + c_3 \frac{1}{T},$$

*for constants* $c_1 = \frac{96K}{\mu\eta^2}$, $c_2 = \frac{24}{\mu\eta^3} + C$ *and* $c_3 = 24(\frac{20}{\mu\eta^2})^2 K + \frac{\mu\eta^2}{2} + C$.

The proof is based on an improvement in the analysis of the UCB Frank-Wolfe algorithm, based on a better control on the duality gap, possible in the strongly convex case. It is a consequence of an inequality due to Lacoste-Julien and Jaggi [24, Lemma 2]. In order to get the result, we adapt these ideas to a case of unknown gradient, with bandit feedback. We note that this approach is similar to the one in [25] that focuses on stochastic optimization problems, as discussed in Section 1.4.

Our framework is more complicated in some aspects than typical settings in stochastic optimization, where strong assumptions can usually be made over the noisy gradient feedback. These include stochastic gradients that are independent unbiased estimates of the true gradient, or with error terms that are decreasing in $t$. Here, such properties do not hold: as an example, in a parametric setting, information is only obtained about one of the coefficients, and there are strong dependencies between successive gradients feedbacks. Dealing with these aspects, as well as the fact that our gradient proxy is penalized by the size of the confidence intervals, are some of the main challenges of the proof.

### 4.3 Interior-smooth functions

Many interesting examples of bandit optimization are not exactly covered by the case of functions that are $C$-smooth on the whole unit simplex. In particular, for several applications, the function diverges at its boundary, as in the examples of Cobb-Douglas utility maximization and variance minimization from Section 1.3. Recall the the loss was defined by

$$\mathbf{E}[\|\hat{\theta} - \theta\|_2^2] = \sum_{i \in [K]} \frac{\sigma_i^2}{T_i} = \frac{1}{T}L(p) = \frac{1}{T}\sum_{i \in [K]} \frac{\sigma_i^2}{p_i}.$$

The gradient Lipschitz constant is infinite but if we knew for instance that $\sigma_i \in [\underline{\sigma}_i, \overline{\sigma}_i]$, we could safely sample first each arm $i$ a linear number of time because $p_i^\star \geq \underline{p}_i := \underline{\sigma}_i/\sum_j \overline{\sigma}_j$. We would have $(p_t)_i \geq \underline{p}_i$ at all stages and our analysis holds with the constant $C = 2\sigma_{\max}^2(\sum_j \overline{\sigma}_j)^3/\underline{\sigma}_{\min}^3$.

Even without knowledge on $\sigma_i^2$, it is possible to quickly have rough estimates, as illustrated by Lemma 2 in the appendix. Only a logarithmic number of sample of each action are needed. Once they are gathered, one can keep sampling each arm a linear number of times, as suggested when the lower/upper bounds are known beforehand. This leads to a Lipchitz constant $C = (9\sum_j \sigma_j)^3/\sigma_{\min}$, which is, up to to a multiplicative factor, the gradient Lipschitz constant at the minimum.

## 5 Lower bounds

The results shown in Sections 3 and 4 exhibit different theoretical guarantees for our algorithm depending on the class of function considered. We discuss here the optimality of these results.

### 5.1 Slow rate lower bound

In Theorem 2, we show a slow rate of order $\sqrt{K\log(T)/T}$ for the error approximation of our algorithm over the class of $C$-smooth convex functions of $\mathbf{R}^K$. Up to the logarithmic term, this result is optimal: no algorithm based on the same feedback can significantly improve the rate of approximation. This is a consequence of the following theorem, a direct corollary of a result by [5].

**Theorem 6.** *For any algorithm based on a bandit feedback such that* $\alpha_{t,i}(T_i, \delta) = \sqrt{2\log(t/\delta)/T_i}$ *and that outputs* $\hat{p}_T$, *we have over the class of linear forms* $\mathcal{L}_K$ *that for some constant* $c > 0$

$$\inf_{\hat{p}_T} \sup_{L \in \mathcal{L}_K} \left\{ \mathbf{E}[L(\hat{p}_T)] - L(p_\star) \right\} \geq c\sqrt{K/T}\,.$$

This result is established over the class of linear functions over the simplex (for which $C = 0$), when the feedback consists of a draw from a distribution with mean $\mu_i$. As mentioned in Section 3, the extra logarithmic term in our upper bound comes from our algorithm, which has the same behavior as UCB. Nevertheless, as mentioned before, modifying our algorithm to recover the behavior of MOSS [4], or even ETC, [see e.g. 29, 30], would improve the upper bound and remove the logarithmic term.

## 5.2 Fast rate lower bound

We have shown that in the case of strongly convex smooth functions, there is an approximation error upper bound of order $(K/\eta^4)\log(T)/T$ for the performance of our algorithm, where $\eta < 1/K$. We provide a lower bound over this class of functions in the following theorem.

**Theorem 7.** *For any algorithm with a bandit feedback such that* $\alpha_{t,i}(T_i, \delta) = \sqrt{2\log(t/\delta)/T_i}$ *and output* $\hat{p}_T$, *we have over the class* $\mathcal{S}_{1,K}$ *of 1-strongly convex functions that for some constant* $c > 0$

$$\inf_{\hat{p}} \sup_{L \in \mathcal{S}_{1,K}} \left\{ \mathbf{E}[L(\hat{p}_T)] - L(p^\star) \right\} \geq c\,K^2/T\,.$$

The proof relies on the complexity of minimizing quadratic functions $\frac{1}{2}\|p - \theta\|_2^2$ when observing a draw from distribution with mean $\theta_i$. Our upper bound is in the best case of order $K^5 \log(T)/T$, as $\eta \leq 1/K$. Understanding more precisely the optimal rate is an interesting venue for future research.

## 5.3 Mixed feedbacks lower bound

In our analysis of this problem, we have only considered settings where the feedback upon choosing action $i$ gives information about the $i$-th coefficient of the gradient. The two following cases show that even in simple settings, our upper bounds will not hold if the relationship between action and feedback is different, when the feedback corresponds to another coefficient.

**Proposition 8.** *For $L$ in the class of 1-strongly convex functions on $\Delta_3$, we have in the case of a mixed bandit feedback that*

$$\inf_{\hat{p}} \sup_{L \in \mathcal{S}_{1,3}} \left\{ \mathbf{E}[L(\hat{p}_T)] - L(p^\star) \right\} \geq c/T^{2/3}\,.$$

For strongly convex functions, even with $K = 3$, there are therefore pathological mixed feedback settings where the error is at least of order $1/T^{2/3}$ instead of $1/T$. The case of smooth convex functions is covered by the existing lower bounds for the problem of *partial monitoring* [13], and gives a lower bound of order $1/T^{1/3}$ instead of $1/\sqrt{T}$.

**Proposition 9.** *For $L$ in the class of linear forms $\mathcal{F}_3$ on $\Delta_3$, with a mixed bandit feedback we have*

$$\inf_{\hat{p}} \sup_{L \in \mathcal{F}_3} \left\{ \mathbf{E}[L(\hat{p}_T)] - L\theta(p^\star) \right\} \geq c/T^{1/3}\,.$$

# 6 Discussion

We study the online minimization of stochastic global loss with a bandit feedback. This is naturally motivated by many applications with a parametric setting, and tradeoffs between exploration and exploitation. The UCB Frank-Wolfe algorithm performs optimally in a generic setting.

The fast rates of convergence obtained for some clases of functions are a significant improvement over the slow rates that hold for smooth convex functions. In bandit-type problems similar to our problem, it is not always possible to leverage additional assumptions such as strong convexity: It has been proved impossible in the closely related setting of online convex optimization [23, 35]. When it

is possible, step sizes must usually depend on the strong convexity parameter, as in gradient descent [28]. This is not the case here, where the step size is fixed by the mechanics of the problem. We have also shown that fast rates are possible without requiring strong convexity, with a gap condition on the gradient at an extreme point, more commonly associated with bandit problems.

We mention that several extensions of our models, motivated by heterogenous estimations, are quite interesting but out of scope. For instance, assume an experimentalist can chose one of $K$ known covariates $X_i$ in order to estimate an unknown $\beta \in \mathbf{R}^K$, and observes $y_t = X_{\pi_t}^\top (\beta + \xi_t)$, where $\xi_t \sim \mathcal{N}(0, \Sigma)$. Variants of that problem with covariates or contexts [29] can also be considered. Assume for instance that $\mu_i(.)$ and $\sigma_i^2(.)$ are regular functions of covariates $\omega \in \mathbf{R}^d$. The objective is to estimate all the functions $\mu_i(.)$.

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
