[Supplementary Material · Supp_BerPer17_camera.pdf]

# Supplementary material to "Fast Rates for Bandit Optimization with Upper-Confidence Frank-Wolfe"

**Quentin Berthet** [*]
University of Cambridge
q.berthet@statslab.cam.ac.uk

**Vianney Perchet** [†]
ENS Paris-Saclay & Criteo Research, Paris
vianney.perchet@normalesup.org

## A  Proofs

**Lemma 1.** *Let $L$ be a $C$-smooth convex function over the unit simplex $\Delta_K$. For any $T \geq 1$, after $T$ steps of the UCB Frank-Wolfe algorithm, it holds that*

$$L(p_T) - L(p_\star) \leq \frac{1}{T} \sum_{t=1}^{T} \varepsilon_t + \frac{C \log(eT)}{T} \,,$$

*where $\varepsilon_{t+1} = (e_{\pi_{t+1}} - e_{\star_{t+1}})^\top \nabla L(p_t)$ is the error compared to Frank-Wolfe with explicit, known and observed gradients, i.e., $e_{\star_{t+1}} = \mathrm{argmax}_{p \in \Delta_K} \, p^\top \nabla L(p_t)$.*

**Remark:** If we denote by $\|L\|_\infty = \sup_{p \in \Delta_K} L(p)$ and $\|\nabla L\|_\infty = \sup_{p \in \Delta_K} \|\nabla L(p)\|$, then the same statements would hold with $(t+1)\|L\|_\infty$ or $2\|\nabla L\|_\infty + \|L\|_\infty$ instead of $\varepsilon_t$.

*Proof of Lemma 1.* We apply the update equation in (1) and follow the usual analysis of Frank-Wolfe in the absence of noise. We denote by $\rho_t$ the approximation error at any time $t$

$$\rho_{t+1} := L(p_{t+1}) - L(p_\star) = L(p_t + \frac{1}{t+1}(e_{\pi_{t+1}} - p_t)) - L(p_\star)$$

By definition of $e_{\star t}$, $C$-smoothness and finally convexity of $L$, we obtain

$$\rho_{t+1} = L(p_t) - L(p_\star) + \frac{1}{t+1}\nabla L(p_t)^\top(e_{\star_{t+1}} - p_t) + \frac{C}{(t+1)^2} + \frac{1}{t+1}\nabla L(p_t)^\top(e_{\pi_{t+1}} - e_{\star_{t+1}})$$

$$\leq (1 - \frac{1}{t+1})\Big[L(p_t) - L(p_\star)\Big] + \frac{1}{t+1}\nabla L(p_t)^\top(e_{\pi_{t+1}} - e_{\star_{t+1}}) + \frac{C}{(t+1)^2}$$

Finally, introducing the notation $\varepsilon_t$ and multiplying by $(t+1)$, we get

$$(t+1)\rho_{t+1} \leq t\rho_t + \varepsilon_{t+1} + \frac{C}{(t+1)} \,.$$

Summing from 1 to $T$ yields the desired result. □

*Proof of Theorem 2.* We use the result of Lemma 1, which yields

$$L(p_T) - L(p_\star) \leq \frac{1}{T} \sum_{t=1}^{T} \varepsilon_t + \frac{C \log(eT)}{T} \,.$$

---

[*]Supported by an Isaac Newton Trust Early Career Support Scheme and by The Alan Turing Institute under the EPSRC grant EP/N510129/1.

[†]Supported by the ANR (grant ANR- 13-JS01-0004-01) and by the *FMJH Program Gaspard Monge in optimization and operations research* (supported in part by EDF) and from the Labex LMH.

We recall that $\varepsilon_t$ is the error due to the lack of information on the gradient at step $t$, and we have

$$\varepsilon_{t+1} := (e_{\pi_{t+1}} - e_{\star_{t+1}})^\top \nabla L(p_t) = \nabla_{\pi_{t+1}} L(p_t) - \nabla_{\star_{t+1}} L(p_t).$$

The difference between the two coefficients of the gradient can be controlled by using the definition of our selection rule, and the relationship between $\nabla L(p_t)$ and $\hat{g}_t$, similarly to the analysis of the UCB algorithm for multi-armed bandit problems. Indeed, by definition of $\hat{g}_t$, we have that with probability at least $1 - \delta_t$, conditionally to the history

$$
\begin{aligned}
\nabla_{\pi_{t+1}} L(p_t) &\leq \hat{g}_{t,\pi_{t+1}} + \alpha_{t,\pi_{t+1}}(T_{\pi_{t+1}}(t), \delta_{t+1}) \\
&\leq \big(\hat{g}_{t,\pi_{t+1}} - \alpha_{t,\pi_{t+1}}(T_{\pi_{t+1}}(t), \delta_{t+1})\big) + 2\alpha_{t,\pi_{t+1}}(T_{\pi_{t+1}}(t), \delta_{t+1}) \\
&\leq \big(\hat{g}_{t,\star_{t+1}} - \alpha_{t,\star_{t+1}}(T_{\pi_{t+1}}(t), \delta_{t+1})\big) + 2\alpha_{t,\pi_{t+1}}(T_{\pi_{t+1}}(t), \delta_{t+1}) \\
&\leq \nabla_{\star_{t+1}} L(p_t) + 2\alpha_{t,\pi_{t+1}}(T_{\pi_{t+1}}(t), \delta_{t+1}),
\end{aligned}
$$

as $\hat{U}_{\pi_{t+1}} \leq \hat{U}_{\star_{t+1}}$ by the definition of our selection rule. With probability $\delta_{t+1}$, we also have that $\varepsilon_{t+1} \leq 2\|\nabla L\|_\infty + \|L\|_\infty$. As a consequence, this yields

$$\mathbf{E}\varepsilon_{t+1} \leq 2\alpha_{t,\pi_{t+1}}(T_{\pi_{t+1}}(t), \delta_{t+1}) + \delta_{t+1}(2\|\nabla L\|_\infty + \|L\|_\infty).$$

We now bound the approximation error as a function of the precision of the estimate $\hat{g}_t$. Using the above inequality, we get, by denoting $\lambda := 2\|\nabla L\|_\infty + \|L\|_\infty$,

$$
\begin{aligned}
\mathbf{E} \sum_{t=1}^{T} \varepsilon_t &\leq K\lambda + \mathbf{E} \sum_{t=K+1}^{T} 2\alpha_{t,\pi_t}(T_{\pi_t}(t-1), \delta_t) + \frac{\lambda}{t^2} \leq 2\mathbf{E} \sum_{t=K+1}^{T} \sqrt{\frac{6\log(t)}{T_{\pi_t}(t-1)}} + \Big(K + \frac{\pi^2}{6}\Big)\lambda \\
&\leq 2\mathbf{E} \sum_{i=1}^{K} \sum_{s=1}^{T_i(T-1)} \sqrt{\frac{6\log(T)}{s}} + \Big(K + \frac{\pi^2}{6}\Big)\lambda \leq 4\mathbf{E} \sum_{i=1}^{K} \sqrt{3T_i(T)\log(T)} + \Big(K + \frac{\pi^2}{6}\Big)\lambda.
\end{aligned}
$$

We used the fact that the algorithm necessarily select actions in a round robin fashion during the first $K$ stages.

Applying Cauchy-Schwarz inequality and the fact that $\sum_i T_i(t) = t$ yield the desired result. $\qquad\square$

*Proof of Proposition 3.* We adapt the proof of Theorem 2, using that $C = 0$ and that $\varepsilon_t = 0$ whenever $\pi_t = \star_t = \star$. We obtained that

$$\mathbf{E}T(L(p_T) - L(p_\star)) \leq 4\mathbf{E} \sum_{i=1}^{K} \sqrt{3T_i(T)\log(T)} + \Big(\frac{\pi^2}{6} + K\Big)\lambda$$

However, in this particular case, we have $T(L(p_T) - L(p_\star)) = \sum_{i \neq \star} \Delta_i T_i$. We therefore obtain

$$\mathbf{E} \sum_{i \neq \star} \Delta_i T_i \leq 4\sqrt{3}\sqrt{\log(T)}\mathbf{E} \sum_{i \neq \star} \sqrt{T_i} + \Big(\frac{\pi^2}{6} + K\Big)\lambda \leq \Big(\sum_{i \neq \star} \frac{48\log(T)}{\Delta_i}\Big)^{1/2} \mathbf{E}\Big(\sum_{i \neq \star} \Delta_i T_i\Big)^{1/2} + \Big(\frac{\pi^2}{6} + K\Big)\lambda,$$

by Cauchy-Schwarz inequality. Standard algebra yields

$$\mathbf{E}L(p_T) - L(p_\star) = \mathbf{E}\frac{1}{T} \sum_{i \neq \star} \Delta_i T_i \leq \frac{48\log(T)}{T} \sum_{i \neq \star} \frac{1}{\Delta_i} + 2\Big(\frac{\pi^2}{6} + K\Big)\lambda$$

$\qquad\square$

*of Proposition 3.* We adapt again the proof of Theorem 2. First of all, notice that $\varepsilon_t \leq 0$ whenever $\pi_t = i_\star$, so that we obtain the following equation

$$\mathbf{E}T\big(L(p_T) - L(p_\star)\big) \leq 4\mathbf{E} \sum_{i \neq i_\star} \sqrt{3T_i(T)\log(T)} + \Big(K + \frac{\pi^2}{6}\Big)\lambda + C\log(eT).$$

Using the fact that $L$ is Lipschitz and that $p_\star = e_{i_\star}$, it also holds that

$$T\big(L(p_T) - L(p_\star)\big) \geq T\big(p_T - p_\star\big)^\top \nabla L(p_\star) = \sum_{i \neq i_\star} T_i \Delta_i(L)$$

Cauchy-Schwartz inequality yields again that

$$\mathbf{E} \sum_{i \neq i_\star} T_i \Delta_i(L) \leq 48 \sum_{i \neq i_\star} \frac{\log(T)}{\Delta_i(L)} + 2\left(K + \frac{\pi^2}{6}\right)\lambda + 2C\log(eT)$$

It remains to lower bound the lhs by the regret. Since $L$ is $C$-smooth, we get also that

$$T\big(L(p_T) - L(p_\star)\big) \leq T\big(p_T - p_\star\big)^\top \nabla L(p_\star) + CT\|p_T - p_\star\|\|^2$$

$$= \sum_{i \neq i_\star} T_i \Delta_i(L) + \frac{C}{T}\sum_{i \neq i^*} T_i^2 + \frac{C}{T}\Big(\sum_{i \neq i^*} T_i\Big)^2$$

$$\leq \sum_{i \neq i_\star} T_i \Delta_i(L) + \frac{CK}{T}\sum_{i \neq i^*} T_i^2 \, .$$

As a consequence, it remains to compute a quantity $\gamma$ such that

$$\mathbf{E} \sum_{i \neq i_\star} T_i \Delta_i(L) + \frac{CK}{T}\sum_{i \neq i^*} T_i^2 \leq \gamma \mathbf{E} \sum_{i \neq i_\star} T_i \Delta_i(L)$$

or at least that for all $i \neq i_\star$

$$\frac{CK}{T}\mathbf{E}T_i^2 \leq (\gamma - 1)\mathbf{E}T_i \Delta_i(L),$$

in particular this is ensured for $\gamma = 1 + \frac{c(1+K)}{\min \Delta_i(L)}$ which gives the result. $\qquad\square$

*Proof of Theorems 5.* Recall that we assumed than on top of being smooth ($C$-Lipschitz gradient), the mapping $L$ is $\mu$-strongly convex and minimized in the relative interior of the simplex. Let $\eta$ be the distance of $p_\star$ to the relative boundary of the simplex then the following Lemma due to ? yields

$$\nabla L(p_t)^\top (p_t - e_{\star_{t+1}}) \geq \sqrt{2\mu\eta^2}\sqrt{L(p_t) - L(p_\star)} \quad \text{and} \quad C \geq \mu\eta^2$$

This implies that

$$L(p_{t+1}) - L(p_\star) = L(p_t) - L(p_\star) + \frac{1}{t+1}\nabla L(p_t)^\top (e_{\star_{t+1}} - p_t) + \frac{C}{(t+1)^2} + \frac{1}{t+1}\nabla L(p_t)^\top (e_{\pi_{t+1}} - e_{\star_{t+1}})$$

$$\leq L(p_t) - L(p_\star) - \frac{\sqrt{2\mu\eta^2}}{t+1}\sqrt{L(p_t) - L(p_\star)} + \frac{C}{(t+1)^2} + \frac{\varepsilon_{t+1}}{t+1}$$

To ease up reading, we introduce the notations, $\alpha = \sqrt{2\mu\eta^2}$ and and $\rho_t = L(p_t) - L(p_\star)$ so that the previous equation rewrites in

$$\rho_{t+1} \leq \rho_t - \alpha\frac{\sqrt{\rho_t}}{t+1} + \frac{C}{(t+1)^2} + \frac{\varepsilon_{t+1}}{t+1},$$

which rewrites again, using the function $\psi(x) = x^2 - \alpha x$, into

$$(t+1)\rho_{t+1} \leq t\rho_t + \Big[\psi(\sqrt{\rho_t}) - \psi(\frac{\varepsilon_{t+1}}{\alpha})\Big] + \frac{\varepsilon_{t+1}^2}{\alpha^2} + \frac{C}{t+1} \, .$$

Recall that we still have the guarantee that $\rho_t \leq \frac{\sum_{s=1}^t \varepsilon_s + \frac{C}{s+1}}{t}$, but we aim at proving some fast rates of convergence, of the type

$$\mathbf{E}\,\rho_T \leq O\Big(\frac{\sum_s \mathbf{E}\varepsilon_s^2}{T}\Big) \, .$$

Assume for the moment that $\rho_T \geq \frac{\alpha^2}{4}$, then Cauchy-Schwarz inequality implies that

$$\Big(\sum_{s=1}^T \varepsilon_s + \frac{C}{s+1}\Big)^2 \leq T\sum_{s=1}^T (\varepsilon_s + \frac{C}{s+1})^2 \leq \frac{4}{\alpha^2}\sum_{s=1}^T (\varepsilon_s + \frac{C}{s+1})\sum_{s=1}^T (\varepsilon + \frac{C}{s+1})^2$$

$$\leq \sum_{s=1}^T (\varepsilon + \frac{C}{s+1})\frac{8}{\alpha^2}\Big(\sum_{s=1}^T \varepsilon_s^2 + \frac{C^2\pi^2}{6}\Big),$$

and thus

$$\rho_T \leq \frac{\sum_{s=1}^{T} \varepsilon_s + \frac{C}{s+1}}{T} \leq \frac{8}{\alpha^2} \frac{\sum_{s=1}^{T} \varepsilon_s^2}{T} + \frac{14C^2}{\alpha^2} \frac{1}{T} \tag{1}$$

As a consequence, the claim holds if $\rho_T \geq \alpha^2/4$ and we will, from now on, assume that $\rho_T \leq \alpha^2/4$.

We denote by $\tau_0$ the last time before $T$ where $\rho_\tau \geq \alpha^2/4$ and we now consider several cases for the remaining of the proof.

**Case1. If we can prove that $\frac{\sum_{s=1}^{t} \varepsilon_s^2}{t} \geq \varepsilon_{t+1}^2$, for example if $\varepsilon_t$ is guaranteed to decrease**

Then, for any $t \geq \tau_0$, we get that if $\rho_t \geq \frac{1}{\alpha^2} \frac{\sum_{s=1}^{t} \varepsilon_s^2}{t} \geq \frac{\varepsilon_{t+1}}{\alpha^2}$ ( by assumption), then

$$(t+1)\rho_{t+1} \leq t\rho_t + \frac{\varepsilon_{t+1}^2}{\alpha^2} + \frac{C}{t+1} \ .$$

Thus, if we denote by $\tau_1$ the last time where $\rho_\tau < \frac{1}{\alpha^2} \frac{\sum_{s=1}^{\tau} \varepsilon_s^2}{\tau}$, we obtain that, as long as $\tau_1 \geq \tau_0$,

$$T\rho_T \leq \tau_1 \rho_{\tau_1} + \varepsilon_{\tau_1+1} + \frac{1}{\alpha^2} \sum_{s=\tau_1+2}^{T} \varepsilon_s^2 + \frac{C}{s+1}$$

$$\leq \frac{1}{\alpha^2} \sum_{s=1}^{T} \varepsilon_s^2 + \varepsilon_{\tau_1+1} - \frac{\varepsilon_{\tau_1+1}^2}{\alpha^2} + C\log(eT)$$

which gives the result we wanted as

$$T\rho_T \leq \frac{1}{\alpha^2} \sum_{s=1}^{T} \varepsilon_s^2 + \frac{\alpha^2}{4} + C\log(eT) \tag{2}$$

On the contrary, if $\tau_0 \geq \tau_1$, then the same computations give

$$T\rho_T \leq \tau_0 \rho_{\tau_0} + \frac{\alpha^2}{4} + \frac{1}{\alpha^2} \sum_{s=\tau_0+1}^{T} \varepsilon_s^2 + \frac{C}{s+1}$$

Using the fact that $\delta_{\tau_0} \geq \alpha^2/4$, we also have that

$$\tau_0 \delta_{\tau_0} \leq \frac{8}{\alpha^2} \sum_{s=1}^{\tau_0} \varepsilon_s^2 + \frac{14C^2}{\alpha^2}$$

thus, combining the two cases $\tau_1 \geq \tau_0$ and $\tau_0 \leq \tau_1$, we now obtain that

$$T\rho_T \leq \frac{8}{\alpha^2} \sum_{s=1}^{T} \varepsilon_s^2 + \frac{14C^2}{\alpha^2} + \frac{\alpha^2}{4} + C\log(eT) \tag{3}$$

**Case 2. If it is not necessarily true that $\frac{\sum_{s=1}^{t} \varepsilon_s^2}{t} \geq \varepsilon_t^2$, for example if $\varepsilon_t$ does not necessarily decrease or can make big jumps**

Notice first that if $\frac{\varepsilon_t^2}{\alpha^2} \leq \rho_t \leq \frac{\alpha^2}{4}$, the latter holding because of $t \geq \tau_0$, then one has

$$(t+1)\rho_{t+1} \leq t\rho_t + \frac{\varepsilon_{t+1}^2}{\alpha^2} + \frac{C}{t+1} \ .$$

As a consequence, denoting by $\tau_2$ the last stage before $T$ such that $\rho_\tau < \frac{\varepsilon_\tau^2}{\alpha^2}$ and assuming that $\tau_2 \geq \tau_0$, we obtain following the same computations as before that

$$t\rho_t \leq \frac{\tau_2 \varepsilon_{\tau_2}^2}{\alpha^2} + \frac{1}{\alpha^2} \sum_{s=\tau_2+1}^{t} \varepsilon_s^2 + \frac{\alpha^2}{4} + C\log(et) \ . \tag{4}$$

If $\tau_2 \leq \tau_0$, then we get that

$$T\rho_T \leq \tau_0 \rho_{\tau_0} + \frac{1}{\alpha^2} \sum_{s=\tau_0+1}^{T} \varepsilon_s^2 + \frac{\alpha^2}{4} + C\log(eT)$$

thus

$$T\rho_T \leq \frac{8}{\alpha^2} \sum_{s=1}^{T} \varepsilon_s^2 + \frac{14C^2}{\alpha^2} + \frac{\alpha^2}{4} + C\log(eT), \tag{5}$$

which was our objective. Hence it only remains to upper-bound $\tau_2 \varepsilon_{\tau_2}^2$ in Equation (4), i.e., when $\tau_2 \geq \tau_0$. To do that, we are going to use a second time the assumptions on $L$.

Since we assumed that $L$ was $\mu$-strongly convex and minimized in the interior of the simplex, it holds that

$$\|p_t - p_*\|^2 \leq \frac{1}{\mu}\big(L(p_t) - L(p_*)\big) \leq \frac{1}{\mu} \frac{\sum_{s=1}^{t} \varepsilon_s}{t}$$

As a consequence, this yields that

$$p_*^i - \sqrt{\frac{1}{\mu}\frac{\sum_{s=1}^{t}\varepsilon_s}{t}} \leq p_t^i \leq p_*^i + \sqrt{\frac{1}{\mu}\frac{\sum_{s=1}^{t}\varepsilon_s}{t}}$$

We are now going to make the assumption that the horizon $T$ is known in advance, and that $\varepsilon_s \leq \left(\frac{\log(T/\delta)}{T_{\pi_s}(s-1)}\right)^{\beta}$ with probability at least $1 - \delta^{\gamma}$, for some $\beta \leq 1/2$ and $\gamma > 0$. This implies, by the union bound, that with probability at least $1 - TK\delta^{\gamma}$,

$$\frac{1}{t}\sum_{s=1}^{t}\varepsilon_s \leq \frac{1}{1-\beta}\Big(\frac{K\log(T/\delta)}{t}\Big)^{\beta},$$

hence

$$tp_*^i - t\sqrt{\frac{1}{\mu}\frac{1}{1-\beta}\Big(\frac{K\log(T/\delta)}{t}\Big)^{\beta}} \leq T_i(t) \leq tp_*^i + t\sqrt{\frac{1}{\mu}\frac{1}{1-\beta}\Big(\frac{K\log(T/\delta)}{t}\Big)^{\beta}}$$

in particular, if $\frac{K\log(T/\delta)}{t} \leq \Big(\mu(1-\beta)\frac{\eta^2}{4}\Big)^{1/\beta}$, i.e., if $t \geq \tau_\delta := \frac{K\log(T/\delta)}{(\mu(1-\beta)\frac{\eta^2}{4})^{1/\beta}}$,

$$\frac{t\delta}{2} \leq \frac{tp_*^i}{2} \leq T_i(t) \leq \frac{3tp_*^i}{2}$$

and thus,

$$t\varepsilon_t^2 \leq t\Big(\frac{\log(T/\delta)}{t\eta/2}\Big)^{2\beta} \leq \Big(\frac{2\log(T/\delta)}{\eta}\Big)^{2\beta} t^{1-2\beta} \leq \Big(\frac{2\log(T/\delta)}{\eta}\Big)^{2\beta} T^{1-2\beta}, \quad \forall t \geq \tau_\delta.$$

**Concluding.**

To wrap things up, we consider the three different cases. With probability at least $1 - TK\delta^{\gamma}$,

**If $\tau_2 \leq \tau_0$ then:** we have proved that

$$T\rho_T \leq \frac{8}{\alpha^2}\sum_{s=1}^{T}\varepsilon_s^2 + \frac{\alpha^2}{4} + C\log(eT)$$

**If $\tau_0 \leq \tau_\delta \leq \tau_2$ then:** using the above upper-bound on $\tau_2\varepsilon_{\tau_2}^2$, we get

$$T\rho_T \leq \frac{1}{\alpha^2}\Big(\frac{2\log(T/\delta)}{\eta}\Big)^{2\beta}T^{1-2\beta} + \frac{1}{\alpha^2}\sum_{s=1}^{T}\varepsilon_s^2 + \frac{\alpha^2}{4} + C\log(eT)\,.$$

**If $\tau_0 \le \tau_2 \le \tau_\delta$ then:** going back to the original induction yields

$$T\rho_T \le \tau_\delta \rho_{\tau_\delta} + \frac{1}{\alpha^2}\sum_{s=1}^{T} \varepsilon_s^2 + \frac{\alpha^2}{4} + C\log(eT) \,.$$

Taking the maximum of all those terms gives that, with probability at least $1 - TK\delta^\gamma$,

$$\rho_T \le \frac{\log(T/\delta)}{T}\frac{K\|L\|_\infty}{(\mu(1-\beta)\frac{\eta^2}{4})^{1/\beta}} + \frac{1}{\alpha^2}\Big(\frac{2\log(T/\delta)}{\eta T}\Big)^{2\beta} + \frac{8}{\alpha^2}\frac{1}{T}\sum_{s=1}^{T}\varepsilon_s^2 + \frac{\alpha^2}{4T} + \frac{C\log(eT)}{T} \quad (6)$$

A simple sommation over $t$ yields that,

$$\frac{1}{T}\sum \varepsilon_s^2 \le \frac{1}{T}\sum \Big(\frac{\log(T/\delta)}{T_i(s)}\Big)^{2\beta} \le \frac{1}{1-2\beta}\Big(\frac{K\log(T/\delta)}{T}\Big)^{2\beta} \qquad \text{if } \beta < \frac{1}{2}$$

and

$$\frac{1}{T}\sum \varepsilon_s^2 \le \frac{1}{T}\sum \frac{\log(T/\delta)}{T_i(s)} \le \frac{K\log(T/\delta)\log(T)}{T} \qquad \text{if } \beta = \frac{1}{2}$$

As a consequence, if $\beta < 1/2$

$$\mathbf{E}\rho_T \le \delta^\gamma TK\|L\|_\infty + \frac{\log(T/\delta)}{T}\frac{K\|L\|_\infty}{(\mu(1-\beta)\frac{\eta^2}{4})^{1/\beta}} + \frac{1}{\alpha^2}\Big(\frac{\log(T/\delta)}{T}\Big)^{2\beta}\Big[\frac{2^{2\beta}}{\eta^{2\beta}} + \frac{8K^{2\beta}}{1-2\beta}\Big] + \frac{\alpha^2}{4T} + \frac{C\log(eT)}{T}$$

and if $\beta = 1/2$

$$\mathbf{E}\rho_T \le \delta^\gamma TK\|L\|_\infty + \frac{\log(T/\delta)}{T}\Big[\frac{K\|L\|_\infty}{(\mu\frac{\eta^2}{8})^2} + \frac{2}{\alpha^2\eta}\Big] + \frac{1}{\alpha^2}\frac{K\log(T/\delta)\log(T)}{T} + \frac{\alpha^2}{4T} + \frac{C\log(eT)}{T}$$

choosing $\delta^\gamma = T^{-(2\beta+1)}$ yields that, if $\beta < \frac{1}{2}$,

$$\mathbf{E}L(p_T) - L(p^\star) \le c_{1,\beta}\frac{\log(T)}{T^{2\beta}} + c_{2,\beta}\Big(\frac{\log(T)}{T}\Big)^{2\beta} + \frac{c_{3,\beta}}{T^{2\beta}}$$

where

$$c_{1,\beta} = \frac{2(\beta+1)K\|L\|_\infty}{\gamma(\mu(1-\beta)\frac{\eta^2}{4})^{1/\beta}} + C, \; c_{2,\beta} = \frac{1}{\alpha^2}\Big(\frac{2(\beta+1)}{\gamma}\Big)^{2\beta}\Big[\frac{2^{2\beta}}{\eta^{2\beta}} + \frac{8K^{2\beta}}{1-2\beta}\Big], \; c_{3,\beta} = K\|L\|_\infty + \frac{\alpha^2}{4} + C.$$

For $\beta = 1/2$, the choice of $\delta^\gamma = T^{-2}$ yields

$$\mathbf{E}L(p_T) - L(p^\star) \le c_1\frac{\log^2(T)}{T} + c_2\frac{\log(T)}{T} + c_3\frac{1}{T}$$

where

$$c_1 = \frac{3K}{\gamma\alpha^2}, \; c_2 = \frac{3}{\gamma\alpha^2}\Big[\frac{K\|L\|_\infty}{(\mu\frac{\eta^2}{8})^2} + \frac{2}{\alpha^2\eta}\Big] + C, \; c_3 = K\|L\|_\infty + \frac{\alpha^2}{4} + C.$$

**Remark:** We assumed that the horizon $T$ was known. If it is not the case, there are two possible ways to deal with that issue to get an anytime algorithm

**Use the Doubling Trick in the algorithm:** The doubling trick is rather classical in online learning, and it consists in running several successive and independent instances of the same algorithm on block of stages of length that increases sufficiently fast enough (so that the error incurred on the first blocks disappears while averaging), but not too fast enough (so that the error during the last block is compensated by the small error cumulated so far on the previous blocks). Its main advantages are that it is simple to describe, to analyze and that it gives the same guarantees of the known horizon, up to some multiplicative constant. The latter depends on the speed of convergence achieved in the known horizon, and it might require careful tuning. The main drawback of the doubling trick is that it regularly discards all the past data and forgets the learning done so far.

In our setting, the correct size of blocks are proportional to $T_j = e^{(\frac{1}{1-\beta})^j}$.

**Use the Doubling Trick in the analysis.** Instead of using the doubling trick in the algorithm, we will prove in the following that we can somehow use it in the analysis of the anytime variant of the algorithm. We first consider the case where $\beta = 1/2$, and we assume that it holds that, for some fixed $\theta > 0$ and for every $s \in \mathbb{N}$, $\varepsilon_s \leq \left(\frac{\theta \log(s)}{T_i(s)}\right)^{\beta}$ with probability at least $1 - \frac{1}{s^6}$.

The immediate consequence of that property is that

$$\frac{1}{T}\mathbf{E}\sum_{s=1}^{T}\varepsilon_s^2 \leq \frac{1}{T}\mathbf{E}\sum_{s=1}^{T}\left(\frac{\theta \log(T)}{T_i(s)}\right)^{2\beta} + \frac{1}{s^6} \leq \frac{2}{1-2\beta}\left(\frac{K\theta \log(T)}{T}\right)^{2\beta}\frac{1}{T}$$

where the last inequality is loose for $\beta < 1/2$ and

$$\frac{1}{T}\mathbf{E}\sum_{s=1}^{T}\varepsilon_s^2 \leq \frac{1}{T}\mathbf{E}\sum_{s=1}^{T}\frac{\theta \log(T)}{T_i(s)} + \frac{1}{s^6} \leq 2\frac{K\theta \log^2(T)}{T}$$

for $\beta = 1/2$.

In order to upper-bound $\mathbf{E}\tau_2\varepsilon_{\tau_2}^2$, we are going to decompose the set of stages in blocks $\mathcal{B}_j = \{t \in [T_{j-1}+1, T_j]\}$ where $T_j = \lfloor e^{(\frac{1}{1-\beta})^j} \rfloor$. As a consequence:

$$\mathbb{P}\left\{\forall k \leq K, \forall s \in \mathcal{B}_j, \varepsilon_s^k \leq \left(\frac{\theta \log(s)}{T_k(s)}\right)^{\beta}\right\} \geq 1 - K\sum_{s \in \mathcal{B}_j}\frac{1}{s^5} =: 1 - p_j.$$

Hence, with probability at least $1 - (p_j + p_{j+1})$, it holds that for all $t \in \mathcal{B}_{j+1}$

$$\frac{1}{t}\sum_{s=1}^{t}\varepsilon_s \leq \frac{T_{j-1}}{t} + \frac{1}{t}\sum_{s=T_{j-1}+1}^{t}\left(\frac{\theta \log(s)}{T_{i_s}(s)}\right)^{\beta} \leq \frac{1}{t^{\beta}} + \frac{2}{1-\beta}\left(\frac{K\theta \log(t)}{t}\right)^{\beta},$$

since $t \geq T_j + 1 \geq T_{j-1}^{\frac{1}{1-\beta}}$.

Following the same argument as in the case where the horizon was known, this implies that with probability at least $1 - (p_i + p_{i+1})$, for all $t \in \mathcal{B}_{i+1}$,

$$T_t^i \geq tp_\star^i - t\sqrt{\frac{1}{\mu}\left(\frac{1}{t^{\beta}} + \frac{2}{1-\beta}\left(\frac{K\theta \log(t)}{t}\right)^{\beta}\right)}.$$

In particular, let $\tau_{\beta,\star}$ be such that $\sqrt{\frac{1}{\mu}\left(\frac{1}{t^{\beta}} + \frac{2}{1-\beta}\left(\frac{K\theta \log(t)}{t}\right)^{\beta}\right)} \leq \frac{\eta}{2}$ for all $t \geq \tau_{\beta,\star}$ and $j_{\beta,\star}$ be the index of the block to which $\tau_{\beta,\star}$ belongs. Then we have that

$$\forall j \geq j_{\beta,\star}, \ \mathbb{P}\left\{\forall t \in \mathcal{B}_{j+1}, t\varepsilon_t^2 \leq \left(\frac{2\theta \log(t)}{\delta}\right)^{2\beta}t^{1-2\beta}\right\} \geq 1 - (p_j + p_{j+1})$$

It follows that

$$\mathbf{E}\tau_2\varepsilon_{\tau_2}^2\mathbf{1}\{\tau_2 \geq T_{j^\star}\} \leq \left(\frac{2\theta \log(t)}{\eta}\right)^{2\beta}t^{1-2\beta} + \sum_{j=j^*}T_{j+1}(p_j+p_{j+1}) \leq 2\left(\frac{2\theta \log(t)}{\eta}\right)^{2\beta}t^{1-2\beta},$$

where, again, the last inequality is loose but compact. This yields the anytime version of the previous theorem, that, for $\beta < 1/2$

$$\forall t \in \mathbb{N}, \ \mathbf{E}L(p_t) - L(p^\star) \leq c'_{1,\beta}\left(\frac{\log(t)}{t}\right)^{2\beta} + c'_{2,\beta}\frac{\log(t)}{t} + c'_{3,\beta}\frac{1}{t}$$

with $c'_{1,\beta} = \frac{2}{\alpha^2}\left(\frac{2\theta}{\eta}\right)^{2\beta} + \frac{8}{\alpha^2}\frac{2}{1-2\beta}(K\theta)^{2\beta}$, $c'_{2,\beta} = C$ and $c'_{3,\beta} = T_{j_{\beta,\star}}\|L\|_\infty + \frac{\alpha^2}{4} + C$.

For $\beta = 1/2$, we get

$$\forall t \in \mathbb{N}, \ \mathbf{E}L(p_t) - L(p^\star) \leq c'_1\frac{\log^2(t)}{t} + c_2\frac{\log(t)}{t} + c_3\frac{1}{t},$$

where $c'_1 = \frac{16K\theta}{\alpha^2}$, $c'_2 = \frac{4\theta}{\eta\alpha^2} + C$ and $c'_3 = T_{j_{1/2,\star}}\|L\|_\infty + \frac{\alpha^2}{4} + C$.

□

**Lemma 2.** *Let $Z_s$, $s \in \{1, \ldots, T\}$ be i.i.d. random variable in $[0,1]$ of expectation $\mathbf{E}Z_s = Z$, then, with probability at least $1 - \delta$, $Z \geq \overline{Z}_\tau/2$ where the random stage $\tau \leq T$ is the first such that $\overline{Z}_\tau \geq \sqrt{2\log(2T/\delta)/\tau}$. As, it also holds that $\overline{Z}_\tau \geq Z - \sqrt{\frac{2\log(T/\delta)}{2t}}$, thus $3\overline{Z}_\tau/2 \geq Z$, we get that*

$$\overline{Z}_\tau/2 \leq Z \leq 3\overline{Z}_\tau/2, \quad \text{for some random} \quad \tau \leq 9\log(2T/\delta)/(2Z^2) + 1$$

This lemma is a direct consequence of Hoeffding's inequality.

*Proof of Theorem 7.* Let $\nu \in (0, 1/29)$, $K > 64\log(2)/\nu$ and $T > 4\nu^2 K^4$. We assume for simplicity that $K$ is even. For $\theta \in \Delta_K$, we consider $L_\theta(p) = \frac{\mu}{2}\|p - \theta\|^2$. We treat first the case of $\mu = 1$. For all $\varepsilon \in \{-1, 1\}^{K/2}$, we consider the vector $\theta_\varepsilon$ such that for all $i \in [K/2]$

$$\theta_{\varepsilon, 2i-1} = \frac{1}{K} + \varepsilon_i \sqrt{\frac{\nu K}{T}} \quad \text{and} \quad \theta_{\varepsilon, 2i} = \frac{1}{K} - \varepsilon_i \sqrt{\frac{\nu K}{T}}.$$

Note that for all $\varepsilon \in \{-1, 1\}^{K/2}$, $p_\varepsilon^\star = \theta_\varepsilon \in \Delta_K$ and that $\nabla L_\theta(p) = p - \theta$, so that an observation from $\mathcal{N}(\theta_i, 1)$ for the $i$-th action constitutes a bandit feedback for the $i$-th coefficient of the gradient with deviation bound $\alpha(T_i, \delta) = \sqrt{2\log(1/\delta)/T_i}$.

Let $\mathcal{M}$ be a subset of $\{-1, 1\}^{K/2}$ such that for all $\varepsilon, \varepsilon' \in \mathcal{M}$, we have $\rho(\varepsilon, \varepsilon') \geq K/8$ and for which $\log(|\mathcal{M}|) \geq K/64$, whose existence is guaranteed by the Varshamov-Gilbert lemma. We have for $\varepsilon, \varepsilon' \in \mathcal{M}$ that $\nu/4 \cdot K^2 T \leq \|\theta_\varepsilon - \theta_{\varepsilon'}\|_2^2 \leq \nu K^2/T$. We consider the subsets $\mathcal{C}_\varepsilon$ of the unit simplex defined by

$$\mathcal{C}_\varepsilon = \left\{ p \in \Delta_K \ : \ \|p - \theta_\varepsilon\|_2^2 < \frac{\nu}{16} \frac{K^2}{T} \right\}.$$

By construction of $\mathcal{M}$, these sets are disjoint.

For any algorithm, on the events where $T_j(T) > 2T/K > T/K + 2\sqrt{\nu K^2 T}$ for some $j \in K$ we have that $\hat{p}_T \notin \mathcal{C}_\varepsilon$. On the events for which $T_j(T) \leq 2T/K$ for all $j \in [K]$, we have that $\hat{p}_T$ can only depend (possibly in a random manner) on an observation from $\mathcal{N}^{\otimes N}(\theta_\varepsilon, I_K)$, where $N \leq 2T/K$. We have that $\mathsf{KL}(\mathcal{N}^{\otimes N}(\theta_\varepsilon, I_K), \mathcal{N}^{\otimes N}(\theta_{\varepsilon'}, I_K)) = N\|\theta_\varepsilon - \theta_{\varepsilon'}\|_2^2 \leq \nu N K^2/T \leq 2\nu K$. Considering together these two events, we obtain as a consequence of Fano's inequality that

$$\inf_{\hat{p}} \max_{\varepsilon \in \mathcal{M}} \mathbf{P}_\varepsilon(\hat{p}_T \notin \mathcal{C}_\varepsilon) \geq 1 - \frac{2\nu K + \log(2)}{K/64} \geq 1 - 129\nu,$$

As a consequence, we have that

$$\inf_{\hat{p}} \max_{\varepsilon \in \mathcal{M}} \left\{ \mathbf{E}[L_{\theta_\varepsilon}(\hat{p}_T)] - L_{\theta_\varepsilon}(p_\varepsilon^\star) \right\} \geq \frac{1}{2}(1 - 129\nu)\frac{\nu}{16} \frac{K^2}{T},$$

which yields the desired result.

□

*Proof of Proposition 9.* For $\theta \in [1/3, 2/3]$, take the class of functions $L_\theta : \mathbf{R}^3 \to \mathbf{R}$

$$L_\theta(p) = \frac{1}{2}(p_1 - \theta)^2 + \frac{1}{2}(p_2 - (1 - \theta))^2 + \frac{1}{2}p_3^2.$$

Consider the case where the mixed feedback for the three actions are drawings from respectively $\mathcal{N}(0, 1), \mathcal{N}(0, 1)$, and $\mathcal{N}(\theta, 1)$. We consider the set

$$\mathcal{C}_\theta = \left\{ p \in \Delta_3 \ : \ \|p - p_\theta^\star\|_2^2 \leq \frac{c}{T^{2/3}} \right\}.$$

For any algorithm, on the event where $T_3(T) > T^{2/3}$, we have $\|p_T - p_\theta^\star\|_2^2 \geq 1/T^{2/3}$ and $p_T \notin \mathcal{C}_\theta$. On the event where $T_3 \leq T^{2/3}$, we have that $\hat{p}_T$ can only depend on a drawing from $\mathcal{N}^{\otimes N}(\theta, 1)$, where $N \leq T^{2/3}$. In this case, we have that

$$\inf_{\hat{p}} \sup_{\theta \in [1/3, 2/3]} \mathbf{E}_\theta[(\hat{p}_{T,1} - \theta)^2] \geq \frac{c'}{N}.$$

Overall this yields the desired result. □