[Reviews · NeurIPS 2017]

Reviewer 1



This paper considers stochastic bandit problem where cumulative loss of player is a global function of actions rather the standard additive linear loss. The authors borrows ideas from Frank-Wolf and UCB algorithms and propose a new algorithm to achieve the optimal regret in this setting, which is quite intriguing Overall, the paper is convincing and well written. The problem is very relevant for machine learning and statistics applications and the proof arguments are quite convincing. I could not find major flaw in the paper, and the theoretical proofs in the paper and appendix look solid to me. However, it would be more convincing if the authors can provide supportive experimental results

Reviewer 2



This is a very interesting paper at the intersection of bandit problems and stochastic optimization. The authors consider the setup where at each time step a decision maker takes an action and gets as a feedback a gradient estimate at the chosen point. The gradient estimate is noisy but the assumption is that we have control over how noisy the gradient estimate is. The challenge for the decision maker is to make a sequence of moves such that the average of all iterates has low error compared to the optima. So the goal is similar to the goal in stochastic optimization, but unlike stochastic optimization (but like bandit problems) one gets to see limited information about the loss function. For the above problem setting the authors introduce an algorithm that combines elements of UCB along with Frank-Wolfe updates. The FW style updates means that we do not play the arm corresponding to the highest UCB, but instead play an arm that is a convex combination of the UCB arm and a probability distribution that comes from historical updates. The authors establish stochastic optimization style bounds for a variety of function classes. The paper is failry well written and makes novel contributions to both stochastic opt and bandit literature. I have a few comments 1. It is not clear to me why the authors consider only Frank-Wolfe algorithm. Is there a deep motivation here or can the arguments made in this paper be translated to even work with other optimization algorithms such as stochastic gradient descent etc...? 2. In lines 214-219, it is not clear to me why one needs to invoke KKT condition. It was not clear to me what the relevance and importance of this paragraph is. Can the authors shine some light here? 3. Definition of strong convexity in line 230 is incorrect. 4. It would perhaps be useful to moved the lines 246-249 to the start of the paper. In this way it will be more clear why and how the problem considered here is different from standard stochastic optimization algorithms. 5. In line 257 what is \sigma_max?

Reviewer 3



The authors consider a problem of sequential optimization over the K-dimensional simplex where, at time t, a learner selects \pi_t \in \{1,...,K\}, and one observes a noisy version of the \pi_t-th component of the gradient of a unknown loss function L evaluated at point p_t = {1 \over t} \sum_{s < t} e_{\pi_s}. The goal is to minimize the final loss E[L(p_T)] after T rounds of such a mechanism. The authors propose and analyze a varient of the Frank-Wolfe algorithm which includes confidence bounds, to account for the uncertainty due to noisy gradient estimates. The authors show a O( \sqrt{K log T \over T} ) regret upper bound for functions L whose gradient is Lipschitz continuous and a O( {K log T \over T} ) regret upper bound in the case where L is strongly convex. Matching lower bounds are also provided. Overall the paper is interesting, the algorithm seems novel and the regret lower and upper bounds match in several cases of interest. My only concern is with the setting which does not seem very general nor intuitive: namely the accuracy of gradient estimates typically increases as time goes by: the error is of order O(1 \over \sqrt{T_i}) where T_i is the number of times i \in \{1,...,K\} has been selected. While this makes sense for the classical multi-armed bandit, this does not seem very natural in most problems (except for the examples provided in the paper, all of them being quite specific). Indeed, the most natural setting seems to be the case where the variance of the noise in the gradient estimate is constant (this is the case considered by most work in stochastic optimization and continuous bandits). Maybe I misunderstood what the authors meant but in the example of sequential allocation shouldnt the variance be \sum_{i=1,...,K: T_i > 0} {\sigma_i \over T_i} and not: \sum_{i=1,...,K} {\sigma_i \over T_i} ? (since the variance must remain bounded)